# Simulated Nitric Acid Rain Aggravated the C and P Limits of Forest Soil Microorganisms

**Meijia Zhou, Jinlong Wang, Haibo Hu *, Jianyu Chen, Ziyi Zhu, Yuchen Heng and Yuanyuan Feng**

Co-Innovation Center of Sustainable Forestry in Southern China, Nanjing Forestry University, Nanjing 210037, China; zmj29287654321@163.com (M.Z.); wjl18751958123@163.com (J.W.); 2180100042@njfu.edu.cn (J.C.); hyc19970411@163.com (Y.H.); feng.yuanyuan@hotmail.com (Y.F.)
* Correspondence: hhb@njfu.com.cn or huhb2000@aliyun.com; Tel.: +86-136-0145-7010

**Abstract:** With the comprehensive emissions of fossil fuel combustion and transportation waste gas, the concentrations of nitrogen oxides ($NO_X$) in the environmental atmosphere increase significantly, leading to nitric acid rain (NAR) pollution. However, the effects of NAR on soil enzyme activities and soil microbial metabolism are unclear. In this study, the *Quercus acutissima* Carruth. forest in the Yangtze River Delta of China was selected as the experimental subject, and was exposed to the simulated spraying of NAR with pH values of 2.5, 3.5, and 4.5 to study the response of the forest soil enzyme activities and soil microbial metabolism to NAR. The results showed that compared to the non-NAR treatment, the activities of β-1,4-glucosidase (BG), L-leucine aminopeptidase (LAP), and β-1,4-N-acetylglucosidase (NAG) decreased by 56.48%–42.24%, 44.57%–38.20%, and 56.13%–48.11% under the AR2.5 and AR3.5 treatments, respectively. Moreover, there was no significant change in the Vector Length (VL) under different gradients of NAR. The Vector Angle (VA) increased with the decrease of the pH value and reached the maximum value with the AR2.5 treatment, indicating that the strong acid type NAR had a greater phosphorus-limiting effect on the soil microorganisms. The RDA analysis results showed that the dissolved organic carbon (DOC) was a significant factor affecting the soil enzyme activity and stoichiometric ratio, with interpretation rates of 40.2%. In conclusion, we believe that in the restoration of acidified soil, attention should be paid to the regulation of soil pH, reducing scour.

**Keywords:** *Quercus acutissima* Carruth. forest soil; acid rain; soil enzyme activities; soil microbial metabolism





## 1. Introduction

Acid rain (AR) refers to the atmospheric precipitation with pH < 5.6 in the atmosphere, and the precipitates ($SO_4^{2-}$, $SO_3^{2-}$, $NO_3^-$, etc.) reaching the surface in the form of precipitation [1,2]. With the comprehensive emissions of fossil fuel combustion and transportation waste gas, the concentrations of sulfur oxides ($SO_X$) and nitrogen oxides ($NO_X$) in the environmental atmosphere increase significantly, leading to AR pollution [3–6]. Soil is the main component of the forest ecosystem, and acid input into the forest soil leads to soil acidification [7]. Some studies have shown that soil acidification will release $Al^{3+}$, which will directly affect the growth of trees, reduce soil microbial activity [8,9], and affect the microbial decomposition of litter, leading to the fundamental cause of forest ecosystem decline [10–12].

Soil enzymes are an important part of the soil ecosystem, a significant index to evaluate the soil fertility and soil microbial activity, and a direct expression of the metabolic needs of the soil microbial community [13,14]. AR can change the soil's physical and chemical properties, thus affecting soil enzyme activities [15]. Similarly, Zhou et al. found that NAR can promote the accumulation of soil nitrogen and inhibit the activity of soil nitrogen metabolism enzymes [16]. Kunito et al. found that soil acid phosphatase activity increased significantly with the increase of the soil pH [17]. However, Lv et al. found that AR had different effects

on different kinds of enzymes [18]. These studies indicate that AR catalyzes the biogeo-chemical process in soil and causes soil acidification or soil nutrient leaching by affecting soil enzyme activity, which has a negative impact on the forest ecosystem. The phosphorus cycling enzyme acid phosphatase (ACP), carbon cycling enzymes β-1,4-glucosidase (BG), and nitrogen-metabolizing enzymes β-1,4-*N*-acetylglucosidase (NAG) and L-leucine aminopep-tidase (LAP) can catalyze the generation of bioavailable end monomers, which are closely related to nutrient cycling in the forest ecosystem [13,19,20]. Therefore, these four enzymes are often used as research subjects, worthy of further study [21].

As reported by Sinsabaugh et al., the relative activities of soil C, N, and P metab-olizing enzymes, namely lnBG: ln (NAG + LAP): ln ACP, were approximately 1:1:1 on a global scale [22]. Additionally, Moorhead et al. proposed the use of VL and VA of enzyme stoichiometry to quantify the relative C, N, and P limits of soil microorganisms, respectively [23,24]. In view of the significant influence of AR on soil enzyme activities, we believe that the use of these two methods can help us intuitively understand the influence of AR on soil enzyme activities and soil microbial metabolism. These methods can reveal the response of the soil enzyme activity and soil microbial nutrient restriction to NAR, and provide reliable suggestions for soil remediation in NAR-polluted areas.

Zheng et al. found that AR can reduce the soil nutrient content and inhibit the soil enzyme activity [15]. However, the method of soil enzyme chemical carrier theory has not been used in the current research on AR. Thus, we simulated the NAR spraying experiment (pH = 4.5, 3.5, and 2.5) in a North subtropical *Quercus acutissima* Carruth. forest. The purpose of our study was to provide suggestions for the restoration of the forest soil damaged by nitric acid rain (NAR). We hypothesized that: (1) NAR would affect the physical and chemical properties of the soil; (2) NAR would have an inhibitory effect on the soil enzyme activity; (3) soil pH would be a key factor affecting the soil enzyme activity and microbial metabolism limitation.

## 2. Materials and Methods

### 2.1. Study Site

This research area was located in the Yangtze River Delta National Forest Ecological Positioning Research Station in Jurong City, Jiangsu Province (Figure 1). The average annual precipitation is 1184.3 mm and the average annual temperature is 15.1 °C, which is a typical north subtropical monsoon climate [18,25]. The soil types are mainly yellow–brown soil and mountain yellow–brown soil, with a pH value of 4.0~5.0. In the experiment site, the natural secondary forest was mainly *Q. acutissima* [16].

### 2.2. Experimental Treatments

In total, 12 sample plots (3 m × 3 m) were set up in the test site on November 2020, and the distance between each plot was 3 m. Four simulated NAR treatments of AR2.5 (pH = 2.5), AR3.5 (pH = 3.5), AR4.5 (pH = 4.5), and CK (pH = 6.5) were randomly conducted. According to the rainfall data of the last 10 years, 2/3 of the average monthly rainfall was determined to be the annual spraying total (Figure 2) [25]. The master batch (0.5 mol/L $H_2SO_4$:0.5 mol/L $HNO_3$ = 1:5) was mixed into tap water to adjust the corresponding pH value. From December 2020 to November 2021, NAR was sprayed on the sample plots in the middle of each month, and an equal amount of tap water (pH = 6.5) was similarly sprayed on the blank plots [16].

### 2.3. Topsoil Sample Collection and Determination

On 30 November 2021, soil samples with 0~10 cm depth from the experimental fields were collected with a circular knife. The stones and plants were separated into three sections after a 2 mm screening. Part of it was immediately used to determine the soil moisture content, part of it was refrigerated at 4 °C for soil enzyme activity analysis, and the rest was air dried naturally for soil chemical properties analysis.

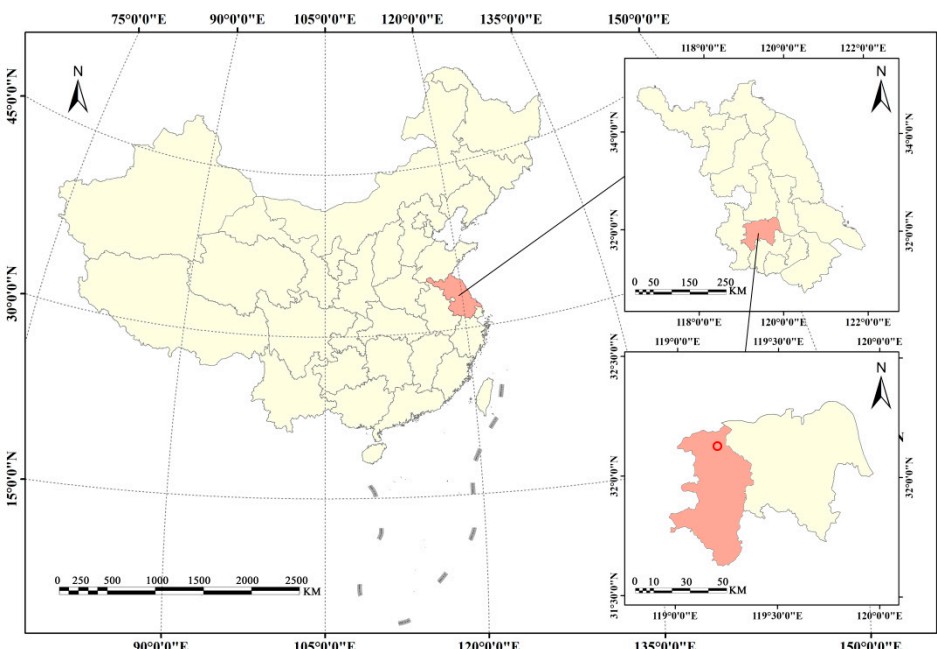

**Figure 1.** Study site.

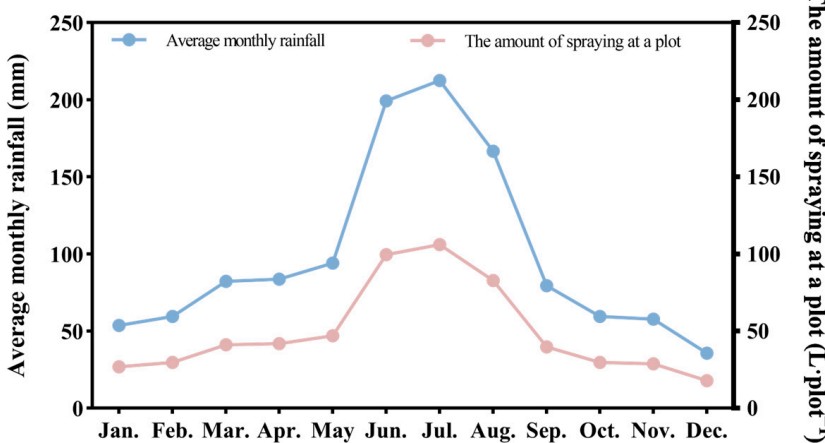

**Figure 2.** Average monthly rainfall and NAR spray volume in the test plots. The spraying amount of a plot = Average monthly rainfall (2011–2021) $\times$ 2/3 $\times$ 1/12 $\times$ 9 m$^2$ (area of a single plot).

The soil water content (SWC) was determined by a drying method. Soil pH was measured by pH potentiometer (soil to water ratio 2.5:1). The soil total nitrogen (TN) was determined by an elemental analyzer (Vario EL III, ELementar, Frankfurt, Germany). The soil organic carbon (SOC) content was determined by $K_2Cr_2O_7$ oxidation-external heating method. The soil total phosphorus (TP) content was determined by molybdenum-antimony resistance colorimetry. The dissolved organic carbon (DOC) content was quantified using a $K_2SO_4$ extraction method, while the soil alkali-hydrolyzable nitrogen (AN) content was measured via an alkaline hydrolysis diffusion technique.

The activities of β-1,4-glucosidase (BG), β-1,4-*N*-acetylglucosidase (NAG), L-leucine aminopeptidase (LAP), and acid phosphatase (ACP) in the soil were tested with the kit measurement test produced by Keming Biology on, (http://www.cominbio.com/index.html (accessed on 3 December 2021)). We quantified the soil microbial nutrient limitation by calculating carrier length and enzyme activity angles for all data based on unconverted proportional activity (e.g., BG/(BG + NAG + LAP)).

$$X = (BG)/(BG + ACP)$$



$$Y = (BG)/(BG + NAG + LAP)$$

$$VL = SQRT(X^2 + Y^2)$$

$$VA = DEGREES[ATAN2(X,Y)]$$

X represented the relative activity of C and P hydrolases, and Y represented the relative activity of C and N hydrolases. The stoichiometric vector analysis of soil enzymes [17] indicated that the longer the stoichiometric relative vector length (VL) of enzymes, the more severe the relative carbon limitation of microorganisms. The relative vector angle of enzyme stoichiometry (VA) > 45° indicated microbial relative phosphorus limitation, <45° indicated microbial relative nitrogen limitation, and the more deviation from the 45° relative limitation, the more severe it became [13].

### 2.4. Analysis Methods

Statistical analysis was used by SPSS software version 21.0 (SPSS Inc., Chicago, IL, USA). One-way ANOVA and multiple comparative analysis (LSD) were used to analyze the differences of the soil chemical properties and enzyme activities ($p < 0.05$). Correlations between the soil chemical properties and enzyme activities was tested using the Pearson correlation analysis. Canoco 5.0 (Microcomputer Power, Ithaca, NY, USA) was used for redundancy analysis (RDA) to reveal the relationship between the soil physical and chemical properties, soil enzyme activities and soil microbial nutrient limitation. GraphPad Prism software version 9 was performed to plot the results.

## 3. Results

### 3.1. Effects of NAR on Soil Chemical Properties

The addition of NAR decreased the soil pH value. Specifically, Table 1 showed that the AR2.5 treatment significantly reduced the soil pH value by 0.17 units relative to the CK treatment ($p < 0.05$). Similarly, the soil total phosphorus (TP) content was significantly reduced by 9.37% only under the treatment of AR2.5 compared to the CK treatment ($p < 0.05$). In contrast, under the AR2.5 treatment, the soil alkali-hydrolyzable nitrogen (AN) content increased significantly by 17.3% ($p < 0.05$). Additionally, when treated with AR2.5 and AR3.5, the soil water content (SWC) was significantly reduced by 9.77% and 16.2%, the soil organic carbon (SOC) content was significantly reduced by 22.4% and 10.3%, and the total nitrogen (TN) content was significantly reduced by 19.6% and 13.1%, compared with the CK treatment (all results $p < 0.05$). Moreover, compared with CK treatment, the NAR treatments significantly reduced the available phosphorus (AP) content by 27.7%, 21.3%, and 14.2%, respectively ($p < 0.05$). The dissolved organic carbon (DOC) exhibited the same rule as AN, and the NAR treatments increased the DOC content by 31.6%, 16.3%, and 12.3%, respectively. Furthermore, the soil SOC/TP (C/P) under the AR2.5 treatment significantly decreased compared with CK, while SOC/TN (C/N) and TN/TP (N/P) had no significant response to NAR ($p > 0.05$).

### 3.2. Changes in Soil Enzyme Activities

As shown in Figure 3, the enzyme activities of β-1,4-glucosidase (BG), L-leucine aminopeptidase (LAP), β-1,4-*N*-acetylglucosidase (NAG), and acid phosphatase (ACP) decreased significantly with the decrease in pH of NAR, and the differences were significant under different NAR treatments. Notably, the activities of BG, LAP, and NAG decreased by 56.48%–42.24%, 44.57%–38.20%, and 56.13%–48.11% under the AR2.5 and AR3.5 treatments, respectively ($p < 0.05$). Moreover, the activities of ACP under the AR3.5 and AR4.5 treatments decreased by 20.83%–20.26% ($p < 0.05$).

**Table 1.** Response of soil chemistry to NAR.

| Treatments | CK | AR2.5 | AR3.5 | AR4.5 |
|---|---|---|---|---|
| pH | 4.53 ± 0.11 a | 4.36 ± 0.06 b | 4.45 ± 0.06 ab | 4.50 ± 0.08 ab |
| SWC (%) | 38.9 ± 0.6 a | 35.1± 4.1 b | 32.6 ± 1.3 b | 38 ± 0.9 a |
| SOC (g·kg$^{-1}$) | 17.4 ± 0.8 a | 13.5 ± 0.7 c | 15.6 ± 0.8 b | 17.4 ± 0.4 a |
| TN (g·kg$^{-1}$) | 1.53 ± 0.06 a | 1.23 ± 0.06 c | 1.33 ± 0.15 bc | 1.47 ± 0.06 ab |
| TP (g·kg$^{-1}$) | 0.32 ± 0.02 a | 0.29 ± 0.01 b | 0.31 ± 0.01 ab | 0.31± 0.02 ab |
| DOC (mg·kg$^{-1}$) | 85.9 ± 10.8 a | 113 ± 4.3 b | 99.9 ± 9.2 b | 96.5 ± 10.9 b |
| AN (mg·kg$^{-1}$) | 110 ± 5 b | 129 ± 9 a | 118 ± 2 b | 114 ± 2 b |
| AP (mg·kg$^{-1}$) | 2.53 ± 0.29 b | 1.83 ± 0.18 a | 1.99 ± 0.05 c | 2.17 ± 0.03 c |
| C/N | 11.4 ± 0.2 a | 11.0 ± 0.2 a | 11.8 ± 0.9 a | 11.9 ± 0.3 a |
| C/P | 54.5 ± 1.1 ab | 46.7 ± 1.9 c | 51.0 ± 3.1 b | 56.2 ± 2.1 a |
| N/P | 4.80 ± 0.17 a | 4.25 ± 0.10 a | 4.35 ± 0.49 a | 4.74 ± 0.23 a |

Data are mean ± standard error. Lowercase letters represent significant differences in soil physical and chemical properties under different NAR treatments ($p < 0.05$).

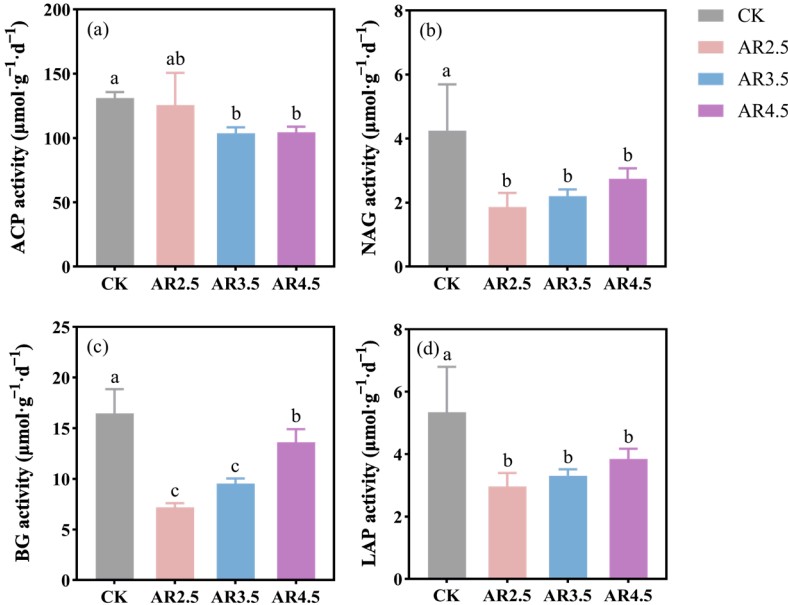

**Figure 3.** Response of soil (**a**) ACP, (**b**) NAG, (**c**) BG, and (**d**) LAP activity to nitric acid rain. Lowercase letters represent significant differences in soil physical and chemical properties under different NAR treatments ($p < 0.05$).

The microbial metabolic limitation was quantified by measuring and calculating the proportion of extracellular enzyme activity (Figure 4). In the NAR treatments, all points (x, y) were located above the diagonal, which meant the P element had a stronger limiting effect. There was no significant change in the VL under different gradients of simulated NAR, compared to the CK treatment. The VAs of the AR2.5 and AR3.5 treatments were higher than that of the CK treatment, while the VA of the AR4.5 treatment had no significant change under NAR stress.

### 3.3. Associations of Microbial C and P Limitation with Other Factors

The BG enzyme activity was positively correlated with the pH (R = 0.811), TN (R = 0.785), SOC (R = 0.836), and N/P (R = 0.742, $p < 0.01$), positively correlated with SWC (R = 0.625) and TP (R = 0.618, $p < 0.05$), and negatively correlated with DOC (R = −0.822) and AN (R = −0.757, $p < 0.01$). Meanwhile, significant consistently negative correlations between the DOC (R = −0.728, $p < 0.01$) and AN (R = −0.675, $p < 0.05$) with the soil NAG were found ($p < 0.01$). Additionally, the activities of LAP were negatively correlated with the DOC (R = −0.688, $p < 0.01$) and AN (R = −0.579, $p < 0.05$). The ACP

enzyme activity was positively correlated with SWC only (R = 0.645, *p* < 0.05) (Table A1). Moreover, Figure 5 shows that the VA was negatively correlated with the pH (*p* = 0.007), TN (*p* = 0.006), SOC (*p* = 0.001) and N/P (*p* = 0.02), while positively correlated with the DOC (*p* = 0.04) and AN (*p* = 0.08).

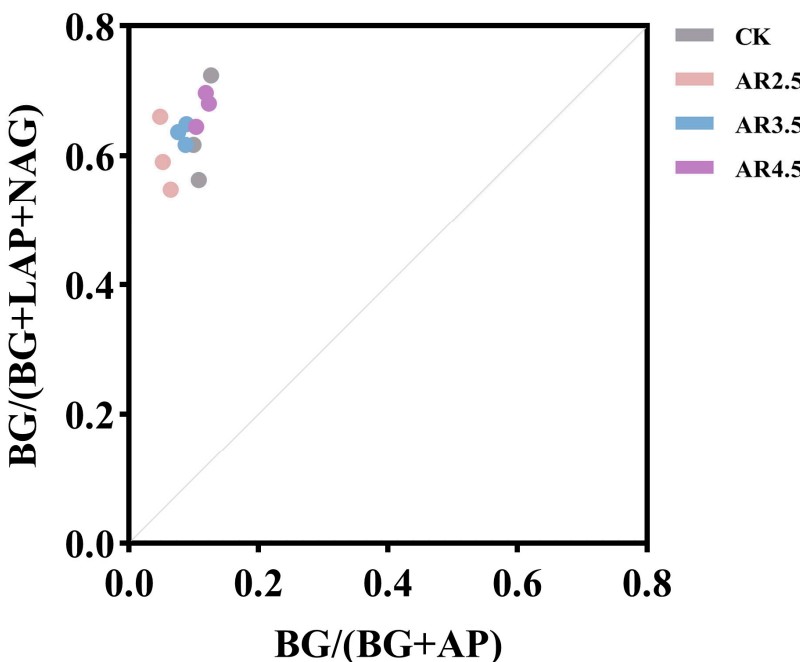

**Figure 4.** The variation of VL and VA.

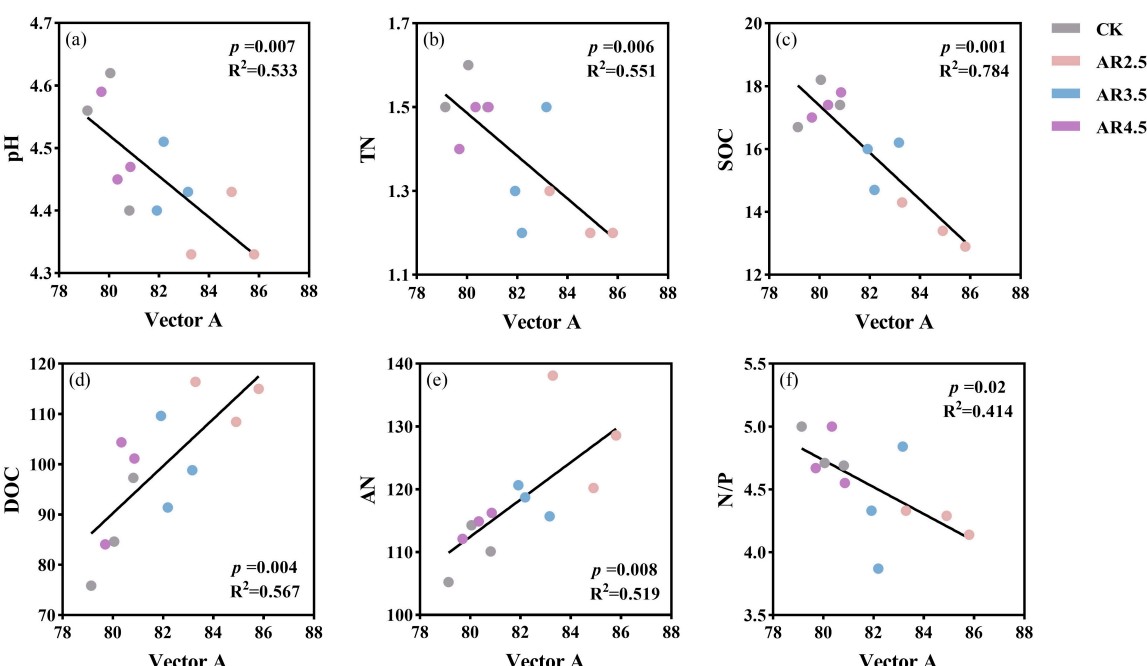

**Figure 5.** Relationships between VA with pH (**a**), TN (**b**), SOC (**c**), DOC (**d**), AN (**e**) and N/P (**f**).

### 3.4. Response of Soil Enzyme Activity and Microbial C, P Limitation to Environmental Factors

In this study, the soil enzyme activity and microbial nutrient limitation were used as response variables, and the soil physical and chemical properties and C, N, and P stoichiometric ratios were used as explanatory variables for the redundancy analysis. Figure 6 shows that the interpretation rate of the first two ranking axes attained 81.81%,

with the first axis explaining 56.85%, and the second axis explaining 24.96% of the variable. Meanwhile, the first axis of the RDA was closely related to AN, SWC and N/P. The second axis of the RDA was closely related to the DOC, VL, AP and ACP. Moreover, the soil DOC ($p$ = 0.002) was a significant factor affecting the soil enzyme activity and stoichiometric ratio, with interpretation rates of 40.2%.

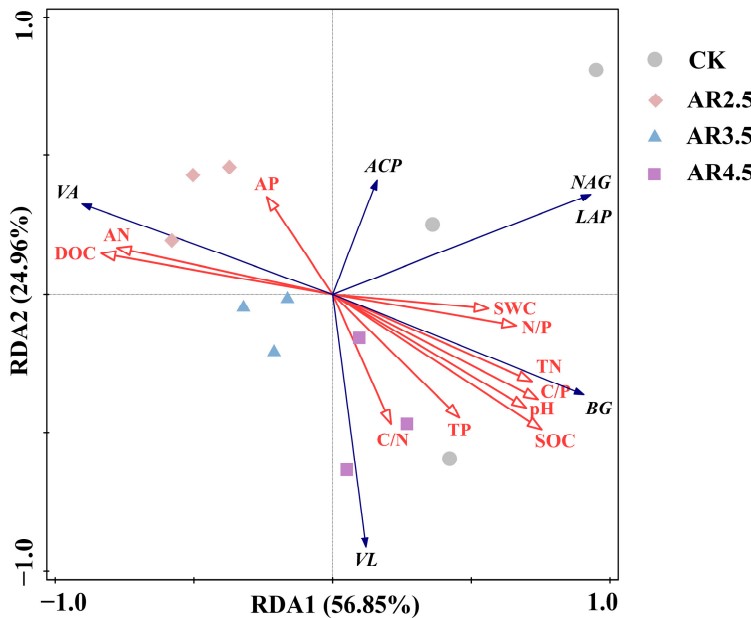

**Figure 6.** Redundancy analysis of soil properties, soil enzyme activities, and microbial C, P limitation.

## 4. Discussion

### 4.1. Effects of Simulated Acid Rain with Different Gradients on Soil Enzyme Activities

Soil enzyme activity can promote soil biochemical reactions and play an important role in maintaining the stability of the forest ecosystem [18]. Some studies [26–28] have proved that AR can cause soil acidification, affect the activity of soil microorganisms, reduce the synthesis of soil enzymes, change the binding mechanism of soil enzymes and humus, affect soil enzyme dissociation, and thus change the activity of soil enzymes. As reported, the soil pH was one of the main factors affecting soil enzyme activity [15,29,30]. In our study, the BG enzyme activity decreased significantly with the decrease of pH of NAR, and the BG enzyme activity was the lowest under the AR2.5 treatment (Table A1), indicating that strong NAR stress would inhibit the soil BG enzyme activity and that the soil pH value was an important factor affecting soil enzyme activity. This is consistent with previous research results [28,29,31]. Additionally, BG was mainly involved in the hydrolysis process of glycosidic bonds between the atomic groups in cellulose, and was related to the relatively unstable carbon pool. The analysis results (Table A1) showed that the BG enzyme activity was significantly positively correlated with the SOC and negatively correlated with the DOC, indicating that the decomposition of the SOC caused by NAR and the change of available C source in the soil were the main factors affecting its activity, which was similar to the results of Chen et al. [32]. Thus, the effect of NAR on the soil enzymes was a complicated process, which was affected by many factors (e.g., soil pH, SOC [32], and soil microbial community [33].

Our study revealed that the activities of NAG and LAP increased with the increase of the NAR concentration. It is generally believed that NAR can inhibit the activity of ammoniating microorganisms in soil, reduce the mineralization rate of soil organic nitrogen, and thus reduce the activity of soil nitrogen metabolism enzymes [34,35]. Moreover, according to the resource allocation theory, abundant inorganic nitrogen sources can reduce the input of soil microorganisms in nitrogen metabolizing enzymes [36], and the activities of NAG and LAP are significantly negatively correlated with AN, indicating that nitrogen

accumulation caused by NAR inhibits the activities of soil nitrogen metabolizing enzymes to a certain extent [15].

Meanwhile, the simulated NAR also inhibited the soil ACP activity in our study. SWC and ACP showed a significant positive correlation; the decrease of ACP activity may be related to the abnormal change of soil SWC [37]. Furthermore, the addition of NAR did not promote the increase of soil moisture, but significantly reduced the soil SWC. As an important index that can directly participate in soil biochemical reactions and affect the life activities of microorganisms and plants, the SWC will directly affect the physiological state of soil microorganisms, limit the ability of microorganisms to decompose some compounds, and regulate soil enzyme activities [19,38]. Meanwhile, the soil pH is also a key factor affecting ACP activity. As reported, soil pH = 5.5 has been proved to be the best pH value for acid phosphatase. However, the soil pH in the study area was significantly lower than 5.5 and was in the trend of acidification, which would inhibit ACP activity [39].

### 4.2. Effects of Simulated Acid Rain with Different Gradients on Soil Microbial Nutrient Limitation

Our study revealed that the VA increased significantly with the decrease in pH of NAR, which corresponded to the significant decrease in the soil AP content, indicating that the simulated AR experiment would promote the loss of the soil P element and strengthen the soil microbial P limitation [21,40]. In addition to reducing the AP content in the soil by inhibiting ACP activity, the simulated NAR may also be because the P element in the north subtropical soil is easily scoured and lost by precipitation [41]. Meanwhile, acid soil is easily combined into a stable closed storage state P, which would reduce the utilization rate of the P element in the soil and aggravate the limitation of the soil microbial P [42]. By transforming the stoichiometric ratio of the soil enzymes in this study area, we found that 1:1.09:1.75 in the CK treatment significantly changed to 1:0.93:2.10 in the simulated NAR treatment, further confirming that the simulated NAR could enhance the C and P limits of the soil microorganisms. These results indicated that the soil microbial inputs to the enzymes involved in soil C, N, and P cycles were significantly different under different concentrations of NAR, which also highlighted the regional characteristics of the subtropical *Q. acutissima* forest [4,43]. Interestingly, the RDA analysis showed that the DOC was an important factor affecting the soil enzyme activity and soil microbial nutrient limitation among the soil physicochemical properties, which was consistent with the research of Wang et al. [44]. This indicated that the soil microbial P limitation was not only related to the availability of the P element, but also affected by the soil DOC content. The DOC played an important role in the regulation of the soil C metabolic enzyme activity and soil microbial nutrient limitation in response to the acid rain treatment. In this study, the soil DOC content increased significantly with AR leaching, which may accelerate the mineralization of forest SOC and soil DOC leaching, resulting in a large loss of wetland SOC [45]. For this reason, the simulated NAR aggravated the soil microbial P limitation, which may not be conducive to the soil organic carbon sequestration and soil carbon cycle in the subtropical *Q. acutissima* forest to a certain extent.

### 5. Conclusions

Nitric acid rain (NAR) caused soil acidification, affected the activity of the soil microorganisms, and thus reduced soil enzyme activity; high concentration NAR had the most significant inhibition effect, BG, NAG, LAP, and ACP all showed this law. Due to the exogenously added NAR, a large amount of $NO_3{}^-$ input played a cumulative role in the soil AN, and also restricted the activity of the nitrogen metabolizing enzymes to a certain extent. It can be seen that the soil pH had a great influence on the enzyme activity. Moreover, the NAR leaching erosion reduced the DOC and AP contents and increased the C and P limits of the soil microorganisms. This may reduce the availability of C for the synthesis of hydrolase to obtain limiting nutrients and further enhanced soil P limits, and it was not conducive to soil DOC retention. In conclusion, we believe that in the restoration of acidified soil, attention should be paid to the regulation of the soil pH, reducing scour.

**Author Contributions:** Formal analysis, M.Z. and J.W.; data curation, J.C., Y.H. and Z.Z.; writing—original draft preparation, M.Z. and J.W.; writing—review and editing Y.F.; supervision, H.H. All authors have read and agreed to the published version of the manuscript.

**Funding:** This research was funded by the special fund project for technology innovation on Carbon Peak Carbon-neutral in 2021, Jiangsu Province, grant number BE2022305; positioning research project of forest ecological system in Yangtze River delta of National Forestry and Grassland Administration, grant number 2021132068 and Technical support and collaboration project from Wuxi Water Conservancy Bureau, grant number 2107116.

**Institutional Review Board Statement:** Not applicable.

**Informed Consent Statement:** Not applicable.

**Data Availability Statement:** The data that support the findings of this study are available from the authors upon reasonable request.

**Conflicts of Interest:** The authors declare no conflict of interest.

**Appendix A**

**Table A1.** Correlation analysis between environmental factors. ** and * indicate significant difference at $p < 0.01$ and $p < 0.05$.

|  | pH | SWC | TN | SOC | TP | DOC | AP | AN | C/N | C/P | N/P |
|---|---|---|---|---|---|---|---|---|---|---|---|
| BG | 0.811 ** | 0.625 * | 0.785 ** | 0.836 ** | 0.618 * | −0.822 ** | −0.201 | −0.757 ** | −0.201 | 0.254 | 0.742 ** |
| NAG | 0.497 | 0.469 | 0.551 | 0.532 | 0.251 | −0.728 ** | −0.155 | −0.675 * | −0.155 | 0.056 | 0.574 |
| LAP | 0.667 | 0.386 | 0.646 | 0.445 | 0.309 | −0.688 ** | −0.121 | −0.579 * | −0.194 | 0.099 | 0.433 |
| ACP | 0.009 | 0.645 * | −0.009 | −0.233 | −0.078 | −0.141 | 0.439 | −0.149 | 0.439 | −0.564 | −0.29 |
| VL | 0.475 | 0.36 | 0.371 | 0.468 | 0.441 | −0.260 | −0.268 | −0.304 | −0.268 | 0.284 | 0.347 |
| VA | −0.730 ** | −0.366 | −0.742 ** | −0.885 ** | −0.534 | 0.766 ** | 0.426 | 0.720 ** | 0.426 | −0.478 | −0.877 ** |

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
