# Peer review of "Simulated Nitric Acid Rain Aggravated the C and P Limits of Forest Soil Microorganisms"

_forests, doi:10.3390/f14051044_

Round 1
Reviewer 1 Report
1. Ideally soil pH is measured in field moist sample, saturation paste or 1:1 soil water suspension. But, in this study, the followed soil to water ratio is 2.5 : 1 for pH measurement of soil. Little explanation is required for this particular ratio.

Author Response
Ideally soil pH is measured in field moist sample, saturation paste or 1:1 soil water suspension. But, in this study, the followed soil to water ratio is 2.5 : 1 for pH measurement of soil. Little explanation is required for this particular ratio.
Response: We appreciate you for your effort to review our manuscript, and your positive feedback. You give an accurate summary of our work and bring forward constructive suggestions. We have made improvements in response to your comments. Thank you for mentioning this point. Based on the reviewer’s comments, the explanation had been deleted to the revised manuscript.
Reviewer 2 Report
The study on "Simulated nitric acid rain aggravated the C and P limits of forest soil microorganisms" is interesting, however the manuscript could be improved with the following remarks:
-change the keyword "nitric acid rain" to another one, since this word is in the title.
-Improve the image quality to 600 dpi, increase the size of the geographic coordinates for better visualization.
-In the statistical analysis section mention that an RDA analysis is going to be performed.
-Improve the description of the results of the RDA multivariate analysis in the results section.
-In the statistical analysis section mention that an RDA analysis multivariate is going to be performed.
Minor editing of English language required
Reviewer 3 Report
I have read the manuscript with the title Simulated nitric acid rain aggravated the C and P limits of forest soil microorganisms and have comments on it. Authors should double-check the text and correct errors. The manuscript cannot be published in its current form. I also find the text inconsistent.
L 43-44 Sentence „As reported, the higher the concentration of AR, the stronger its inhibitory effect on soil enzyme aktivity“ Omit. It doesn't make sense in the text.
L 46 Kunito et al. … the reference year is missing
L 47 Lv et a. … the reference year is missing
L 51 L-leucine aminopeptidase does not metabolize phosphorus.
L 57 What a meta-analysis?
L 58 What is ln AP? Abbreviation explanation missing.
L59 Moorhead et al. … the reference year is missing
L66 Which previous studies?
L69 Which researchers?
L131 Are there any references missing?
Table 1 What is the abbreviation AN?
L165-168 Overwrite chart description.
L201 Which studies?
L217 …many factors…. Which factors?
L227 NAR does not just inhibit ACP. Correct.
L258 Are there any references missing?
Round 2
Reviewer 3 Report
The authors responded to all my comments and revised the text. So I recommend the manuscript for publication.